# Dynamic Vascular Imaging Using Active Breast Thermography

**DOI:** 10.3390/s23063012

**Published:** 2023-03-10

**Authors:** Meir Gershenson, Jonathan Gershenson

**Affiliations:** 1MCM Research Laurel, 134 Cholul YU, Merida 97305, Mexico; 2Mercy Radiology Group, Dignity Health Advanced Imaging, Sacramento, CA 95817, USA

**Keywords:** breast cancer, components analysis, dynamic thermogram, equivalent wave field, vasomodulation, veins, virtual wave

## Abstract

Mammography is considered the gold standard for breast cancer screening and diagnostic imaging; however, there is an unmet clinical need for complementary methods to detect lesions not characterized by mammography. Far-infrared ‘thermogram’ breast imaging can map the skin temperature, and signal inversion with components analysis can be used to identify the mechanisms of thermal image generation of the vasculature using dynamic thermal data. This work focuses on using dynamic infrared breast imaging to identify the thermal response of the stationary vascular system and the physiologic vascular response to a temperature stimulus affected by vasomodulation. The recorded data are analyzed by converting the diffusive heat propagation into a virtual wave and identifying the reflection using component analysis. Clear images of passive thermal reflection and thermal response to vasomodulation were obtained. In our limited data, the magnitude of vasoconstriction appears to depend on the presence of cancer. The authors propose future studies with supporting diagnostic and clinical data that may provide validation of the proposed paradigm.

## 1. Introduction

Breast cancer is one of the most commonly diagnosed and deadly cancers among women. Early detection is crucial to improving outcomes [1]. The most common clinically used modalities for screening and diagnostic breast imaging are mammography, ultrasound, and MRI. Mammography is the gold standard for screening and guidance of tissue sampling for suspected neoplasms. However, certain limitations exist. The sensitivity of mammography to detect tumors decreases with increasing breast density, as breast parenchyma’s attenuation coefficient versus tumor is closer in value than adipose.

Additionally, modest concern exists for an expectation of mammography to result in the development of iatrogenic-related malignancies among the breast cancer screening population. In the global crisis, mammography remains cost-prohibitive for implementation in developing countries. Unmet clinical needs exist for additional and complementary methods to address the current limitations of breast cancer imaging and further address the global issue of breast cancer. Thermography is an emission scan of medium or long-wave infrared (I.R.) electromagnetic radiation emitted via the body, depending on temperature. Thus, the I.R. image of the body is equivalent to an overall map of the external temperature of individual pixels [2]. A sensitivity of 0.2 °C is easily achieved in OEM products [3]. As the emissivity of the skin is almost uniform, the temperature sensitivity realized over the skin surface is unaltered. While thermography was approved by the FDA as an adjunct method to mammography in 1982, its utility is minimal due to low sensitivity, poor ability to distinguish between healthy and diseased tissue, and lack of large population studies. As a result, the FDA issued a warning about using thermography for breast cancer screening [4]. Characteristically, cancer is generally distinguished by an increased rate of metabolism, in part responsible for temperature increase [5].

Malignancy is characterized by increased blood supply via angiogenesis and regulatory pathways [6,7]. Penne’s bioheat transfer equation [8] describes the thermal propagation in live tissue. In vivo measurements establishing tumor hyperthermia were shown by Gautherie [9], observing a typical 1.5 °C temperature difference between the tissue and the average temperature of the blood or, similarly, the temperature difference between the incoming blood to the outgoing blood. It is a consensus among all the modeling based on the bioheat equation [10,11] that the blood perfusion absorbs most of the heat generated by the tumor. None of the models explain what happens to the heat that is carried away by the veins. This heat dominates the appearance of the diseased breast. As a result, the heat signature appears away from the cancerous tissue, where the blood flows superficially towards the skin surface and dissipates to the surrounding environment. The temperature of the blood pool within the arterial vascular system defines the body core temperature’s assumedly uniform distribution throughout the body [12]. Thus, we can detect areas of higher heat signal revealed as a contrast within the image, most evident as ‘hot veins’ carrying increased tumor heat. No publications show a correspondence between vasculature heat signature localization and anatomical tumor location. The use of I.R. imaging to identify neovascularization has some utility. Novel thermal imaging techniques use what is called “thermal challenge,” “cold-stress,” or “dynamic imaging,” where either the breast or some other part of the body is cooled while taking the thermogram; in Y Ohashi’s work [13] subtracting the cold series images from the initial images taken before cold stress “diagnostic accuracy improved from 54% in steady-state thermography to 82% in dynamic” thermography. In previous work [14,15], the authors showed the use of virtual wave transform (VWT) applied to dynamic data to detect increased perfusion associated with the tumor.

VWT was developed as a nondestructive testing tool and has successfully solved the ill-posed, inverse problem related to heat transfer [16]. VWT mathematically can be described as a linear transform that converts thermal diffusive propagation into associated media with similar geometry that supports wave propagation. We introduce a supplementary study of VWT in Appendix B. Yousefi et al. [17,18,19] employed PCA in the dynamic data to preprocess vein data for further artificial intelligence (A.I.) interpretation. The most common methods used in interpreting breast thermograms are inverse thermal modeling and identification of neovascularization, creating new blood vessels in response to increased metabolic activity. Lately, there has been much recent research into thermal breast modeling with improved geometrical details of the breast [20,21,22]; the ability to distinguish between an internal heat source and heat from superficial blood flow will benefit the methods. The emergence of A.I. has resulted in a large number of papers. The Karolinska institute [23] conducted an early study using A.I. with an extensive database; it included 1727 women characterized as having dense breasts. Mammography detected seven cancers, while thermography detected six additional cases but missed all but one of those detected by mammography. Those are six cases not detected by mammography. Later, smaller A.I. studies claim better statistics [24]. Thermography may be used in diagnosing breast cancer and treatment, especially in estimating radiation reactions after radiotherapy [25]. For thermal imaging as breast cancer screening, we refer the reader to review articles [26,27,28,29].

## 2. Method

### 2.1. Goal

The overall goal was to identify the heat generated by the tumor. The problem is that most of the internal heat is carried away by the veins. The heat dissipating through the venous system obscures the location of the heat source. In this work, we had a limited goal of removing the veins’ heat, so we could later identify the cancer self-heat. Later, during this work, we recognize the thermal response and the images of the vasoconstriction of the veins. We should stress that the scope of this work is limited to algorithm development, as the data quality was insufficient to make medical conclusions.

### 2.2. Approach

We used dynamic image data collected following external cooling of the skin with airflow by a fan. We analyzed the response to external cooling to investigate and remove the contribution of the veins from the thermogram. The response of the veins to cooling contains two parts active and passive components. The passive response is similar to typical solid materials. In non-biologic material, heat propagation follows the diffusion equation. Heat is reflected depending on thermal impedance mismatch at boundaries. We observe total reflection at the interface with an insulator, similar to thermal nondestructive evaluation (NDE). Total negative reflection happens at boundaries held at a constant temperature; vasculatures with large blood flow are such boundaries. The other response is unique to the vasculature system: When applying external temperature change, to maintain constant body core temperature, the physiological response to cooling by the venous system is vasoconstriction. This response comprises two processes, a local one controlled with a direct neural connection from the skin to the vasculature and a remote one controlled by the hormonal secretion of norepinephrine [30]. The first response is almost immediate, while the second one is delayed. The vasomodulation response is at the same polarity as the external stimulus. When we cool the skin, the vasculature contraction reduces the released heat. The regular reflections from vasculature held at constant temperature are different. The boundary condition of constant temperature at the interface of the vasculature dictates a backward propagating signal of reverse polarity of the incoming signal. Identifying those responses will enable us to identify and isolate the vasculature from the remaining thermal signal. 

### 2.3. Mathematical Analysis

We borrow techniques used in thermal NDE. The 21 individual frames (20 time-lapse and a static one) are converted into 21 vectors, becoming a single matrix of 21 columns. We keep a tab on the correspondence between vectors and image indices. We use two mathematical tools to analyze the data, VWT (Appendix B) and matrix factorization (Appendix C). By applying VWT, we perform time focusing on the images. It is equivalent to a matrix factorization ***I = T M*** where ***I*** is the original image, and the time matrix ***T*** is ***W*** of Equation (A8). The image matrix ***M*** becomes a virtual wave with wave-like properties; each column represents an image at a different depth. The transformation is simple as it requires only matrix multiplication. The matrix ***W*** has no information unless we want to reconstruct the original data. Performing VWT also rejects images that do not follow diffusion propagation in response to the cooling. As the transformation is ill-posed, time focusing is imperfect, and images in matrix ***M*** still overlap. We remedy the overlap in two stages; in the first step, we use PCA. PCA factors the converted matrix based on similarities between images; it cannot separate similar-looking images that arrive at different times. Next, we separate the images based on their arrival time by applying ICA to the resulting time matrix. The process uses three alternate matrix factorizations; time-based and image-based, followed by a time-based. We give more details in the examples.

## 3. Results

### 3.1. Data

To obtain the data from the Brazilian Database for Breast Research [31], we used publicly accessible methods. This data of opportunity was collected without a specific analysis technique in mind. The data are problematic [32]. Patients are classified only as “sick” or “healthy” with no supporting information about the diagnosis. Only four patients classified as “sick” had either the left or right breast identified. During the collection process, the room temperature was kept between 20 °C to 22 °C, and a fan was used to cool the skin surface until it reached 32.5 °C for no more than 5 min. The recording started once the cooling process was complete, and data were sampled every 15 s for 5 min. The patient stood in front of the camera without support or stabilization. Motion artifacts were a significant limitation affecting many studies examined in the data set. For details of the procedure, the reader is referred to the original publication [31]. Due to the limitations of the data, we used it only to develop the algorithms but did not make any medical conclusions. We analyzed only four optimal cases, with the laterality of the diseased breast provided with the data. We present patients #T281 and #T285; in the Appendix A, we present patients #T282 and #T286. In Figure 1, we display the static images. Dynamic images are almost identical to static ones. Medical data were minimal; Table 1 summarizes them. Some patients were characterized only as “healthy” or “sick” with no additional details.

### 3.2. Data Preparation

First, motion correction methods were applied (Appendix E). Second, we subtracted from each image 21 °C representing the background room temperature. We segmented the left and right breasts for individual analysis. A matrix represents each image, and the matrixes were converted into a vector and stacked to form a single matrix. We included the static image as the final 21st frame. We included two sets of data: a masked version of the images with only pixels from regions of the segmented breast and a second with the nearby surrounding pixels. We performed the segmentation manually. We calculated the transforms using the segmented masks and applied them to the masked data for further analysis, and we applied them to the unmasked data for display. Using the masked data we extract the excitation profile of each patient (Appendix D)

### 3.3. Analysis

First presented is patient #T281; the right breast is labeled “healthy.” After data preparation, we applied VWT as described in Appendix B. Parameters used for the transform: Hermit polynomial order *n* = 2 and regularization parameter *λ = 0.003.* For the diffusivity, ‘α’ of subcutaneous fat 7.58 × 10^−8^ m^2^s^−1^. We applied PCA next using six components with 100 virtual time points at the sequence *t_i_* = 0.025 × 2^(*i −* 100)/20^. One hundred points are more than necessary but result in smoother tracings. We show the images in Figure 2, images 1 to 6. We present the amplitudes of the columns of the time matrix in Figure 3. The waveforms of different patients and contralateral breasts are very similar. On the log time scale, columns one and two appear similar but are shifted. In these traces, we recognized first a small negative peak, then a larger positive peak at twice the time of the first peak. The PCA algorithm separates the components based on a similar appearance but cannot distinguish between almost identical images at different depths. Tracings of Figure 2, column 4 peaks are collocated with the peaks of column 2, but in column 4, both are positive. We interpret it as image 2 being composed of two very similar images, while image 4 is the difference between them. The traces of columns 1 and 3 are very similar to those of 2 and 4 but less pronounced. The original application of PCA to the images separates similar images but mixes the time. We unmixed the components by applying ICA to the time matrix. At the same time, we apply the inverse to the corresponding image to maintain the product matrix. Block diagram of the processing is depicted in Figure 4. We show the first four traces of the unmixed time matrix in Figure 5 and the images in Figure 6. Assuming a fat layer, the estimated depth of trace 1 is 13 mm deep and 9 mm for trace 2. 

By applying the inverse transformation to images 3-1 to 3-4, we get the images in Figure 6. In Figure 7, we present the result for the left breast, labeled “sick”; for patient #T285 in Figure 8(1–4) for the left breast, labeled “healthy”; and for the right breast, labeled “sick,” in Figure 9. The Appendix A presents process images of patients #282 and #286. Results for all the four patient are posted in the Appendix A.

## 4. Discussion

Breast imaging is a multi-modality specialty with screening performed in series and parallel diagnostic testing performed to provide additional capability for the breast radiologist. The authors recognize a single most clinically significant and applicable study using thermography to address the current unmet clinical need: identifying additional breast cancers in patients with dense breasts not detected by mammography. The authors’ motivation for this work is to enable image interpretation beyond statistically based prognostic estimation. We identified two very similar thermal images at each breast. The first image is at the same polarity as the external cooling. The second image is from twice the time of the first one, corresponding to twice the depth and at the opposite polarity. Our interpretation is: The first reflection is due to vasoconstriction, while the second is due to thermal reflection. Skins nerves trigger vasoconstriction, which results in almost instantaneous constriction. The response to external cooling as a thermal regulation mechanism is the reduction of the heat conduction out of the vascular system, which is at the same polarity as the external cooling. The heat propagates only one way from the vascular system to the skin. The response by thermal reflection is different; the cooling traverses from the skin to the blood vessel, then reaches a constant temperature boundary, blood flows, and propagates back to the skin. The outcome is a signal with the reverse polarity of the cooling at twice the time of the first image. Figure 6, Figure 7, Figure 8 and Figure 9 show reflected heat in images 1 and 3, while images 2 and 4 show vasoconstriction-generated signals. Comparing the images of the healthy and sick breast, we notice that the two images of reflection and vasoconstriction of the healthy breast are almost identical; on the other hand, in the sick breast, of Figure 7 and Figure 9, the vasoconstricted images 1 and 2 are different from the reflected signal of images 2 and 3. Some vessels in the vasoconstricted images are not visible or are at reduced intensity; they barely constrict compared to a healthy one. It is visible in patient T285 and inpatient T286, and the difference is visible on the upper side quadrant, the same as the patient diagnosis. We noticed similar behavior in the Appendix A. Yousef [17] suggested using vasodilation as a breast cancer screening method. He used the same data set as this study. By using matrix factorization, he showed that he could improve cancer screening, attributing it to vasodilation.

## 5. Conclusions

A lack of evidence and interpretable data has caused a 40-year barrier in the clinical adoption of an FDA-approved imaging modality for breast radiologists. The authors present a new method for actual patient data, providing visual correspondence. The initial goal of this work was to isolate the blood vessels as preprocessing to identify deeper, static heat sources based on thermal reflection by the constant temperature of the blood vessels. During this work, we identified a second mechanism of thermal reflection by vasoconstriction. The limited but valuable data support the hypothesis that the magnitude of vasoconstriction is detectable in dynamic images related to the presence of cancer. Only 20 frames were collected in the data used, while the camera can obtain 900 within the same period. A dedicated protocol using a high data rate would result in a 30-time signal-to-noise ratio improvement and improved ill-posed inversion performance. In such cases, PCA and ICA processing might be optional.

Additionally, the application of the 3D algorithm is enabled in this way. Considerably, much recent research focuses on improved modeling with geometrical details [20,21,22]. The authors highlight a fundamental unaddressed issue: cancer may be unresolved or obscured due to vascularity which is intrinsic to imaging in the long-wave infrared spectrum. The scope of this work is limited to algorithm development; future studies applying this work are needed to provide statistics for diagnostic testing.

## Figures and Tables

**Figure 1 sensors-23-03012-f001:**
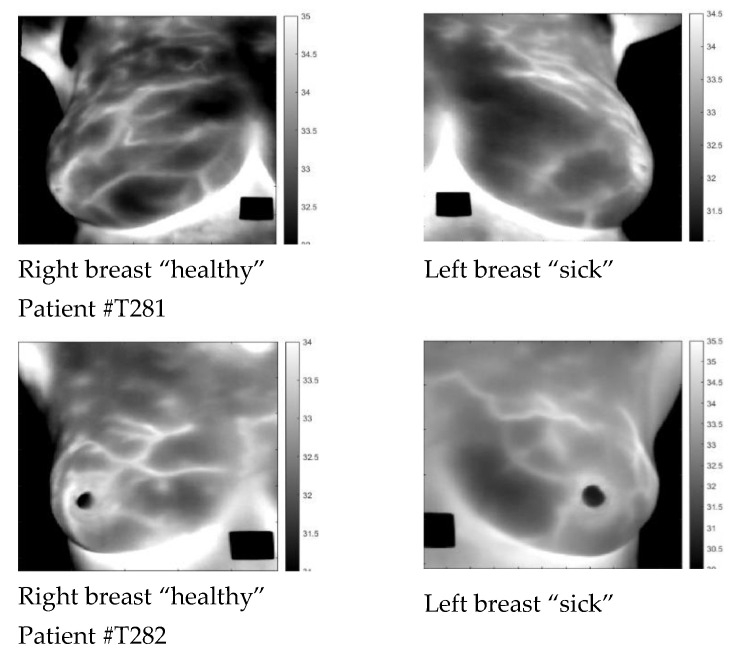
Raw static images of patients #T281 and #T282.

**Figure 2 sensors-23-03012-f002:**
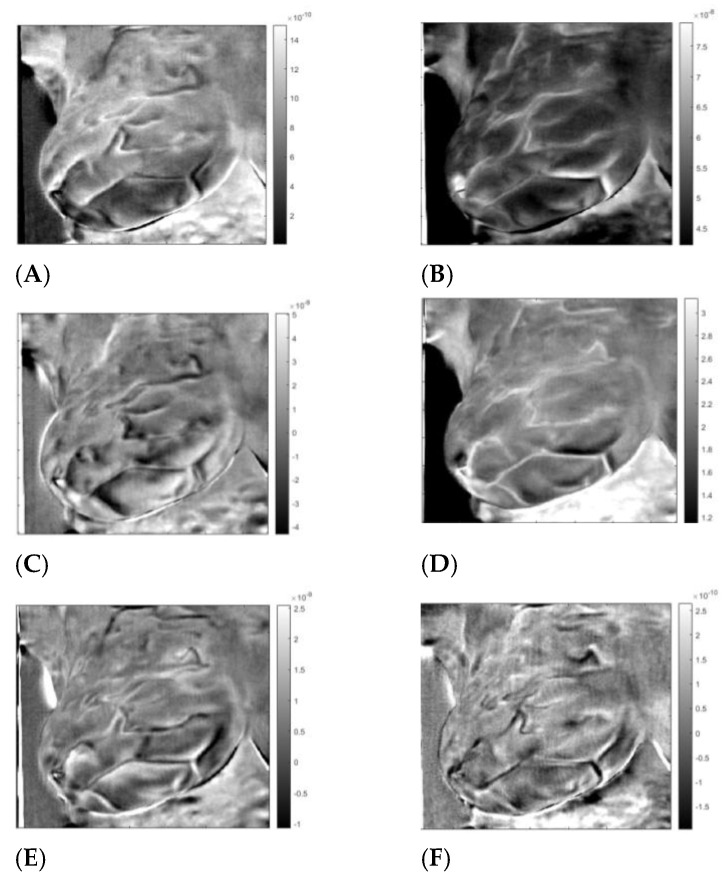
Thermal images of patient “T281”; right breast after VWT and PCA. Images A to F are individuzal PCA vectors converted to images (**A**–**F**).

**Figure 3 sensors-23-03012-f003:**
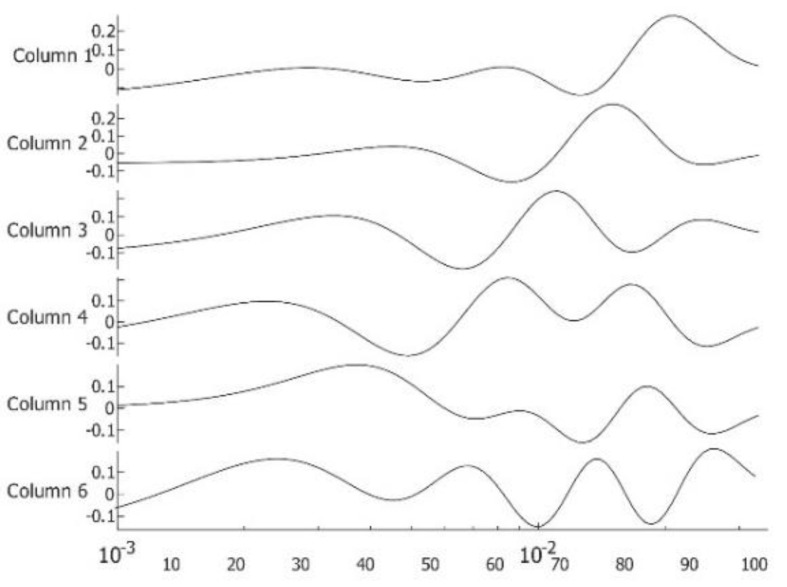
Amplitude of images 1–6 in Figure 2 over virtual time on log time scale with the sequence *t_i_* = 0.025 × 2*^(i −^*
^100*)*/20^. Traces 1–6 are time amplitudes of Figure 2A–F.

**Figure 4 sensors-23-03012-f004:**
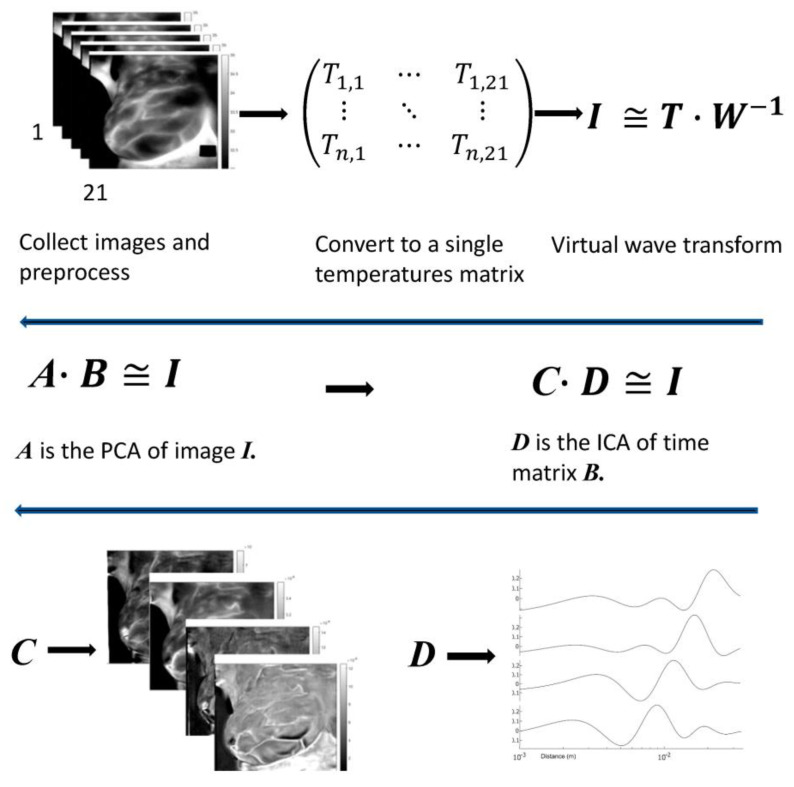
A simplified block diagram of the processing. *n* is a count of all the pixels included in the individual breast. ***A*** and ***C*** are image matrixes, and ***B*** and ***D*** are time matrixes. In the example given, matrix ***I*** is *n* × 100, ***A*** dimensions are 100 × 6, ***B*** dimensions are 6 × *n*, ***C*** dimensions are *n* × 4, and ***D*** dimensions are 4 × 100.

**Figure 5 sensors-23-03012-f005:**
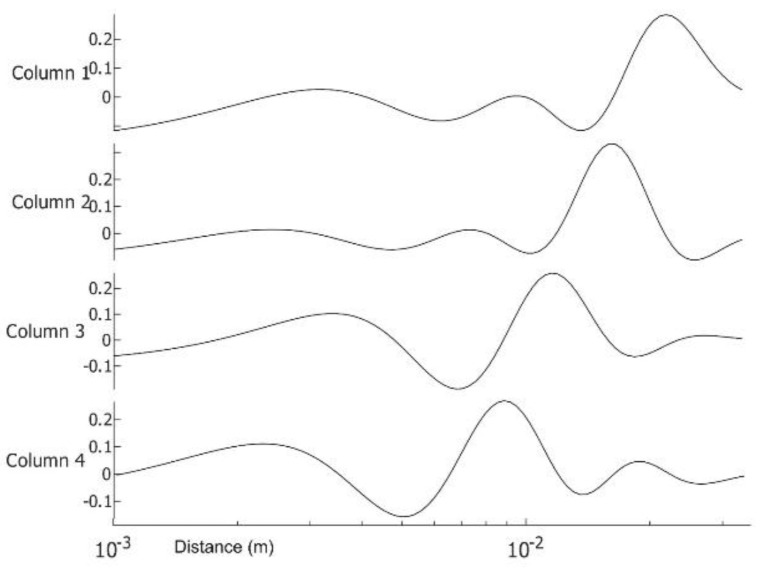
Traces 1:4 after application of ICA to the time matrix of Figure 3 on log time scale with the sequence *t_i_* = 0.025 × 2^(*i −* 100)/20^.

**Figure 6 sensors-23-03012-f006:**
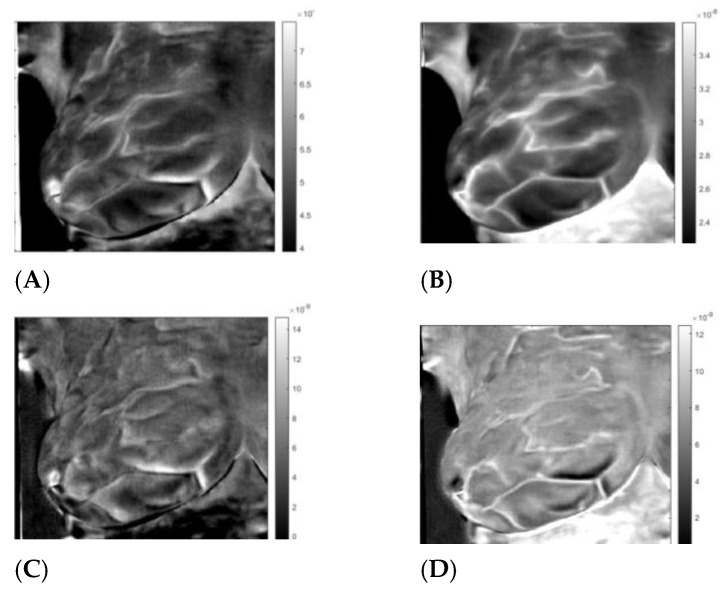
Healthy right breast of patient “T281.” Images 1A to 1D after application of the inverse ICA transform. (**A**,**B**) are images of thermal reflection, while (**C**,**D**) are images of vasoconstriction.

**Figure 7 sensors-23-03012-f007:**
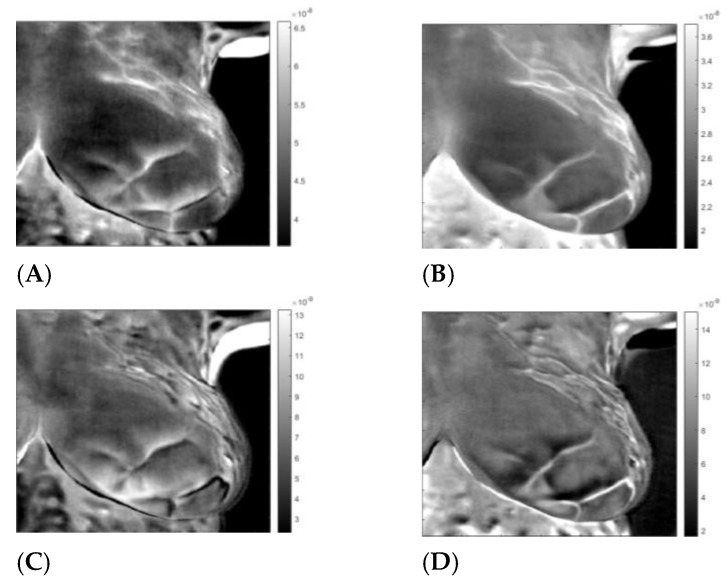
Sick left breast of patient “T281.” Images A to D after application of the inverse ICA transform. (**A**,**B**) are images of thermal reflection, while (**C**,**D**) are images of vasoconstriction.

**Figure 8 sensors-23-03012-f008:**
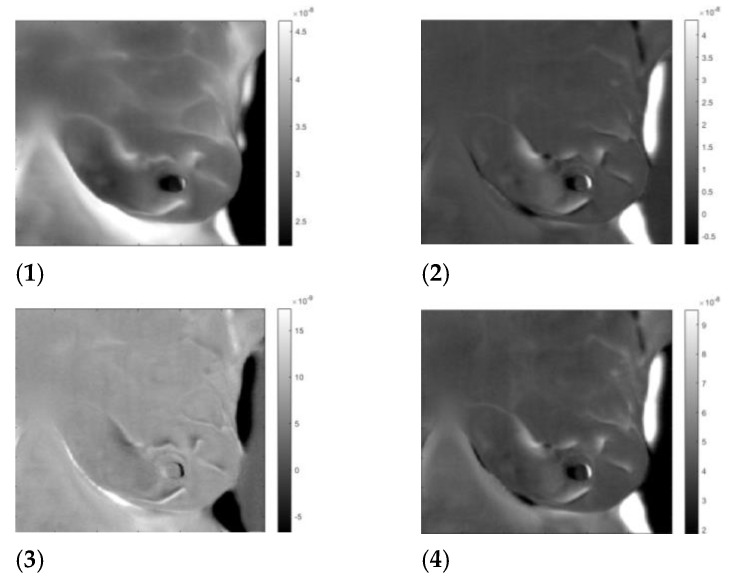
“Healthy” left breast of patient #T285. Images 1 to 4 after application of the inverse ICA transform. (**1**) and (**2**) are images of thermal reflection, while (**3**) and (**4**) are images of vasoconstriction.

**Figure 9 sensors-23-03012-f009:**
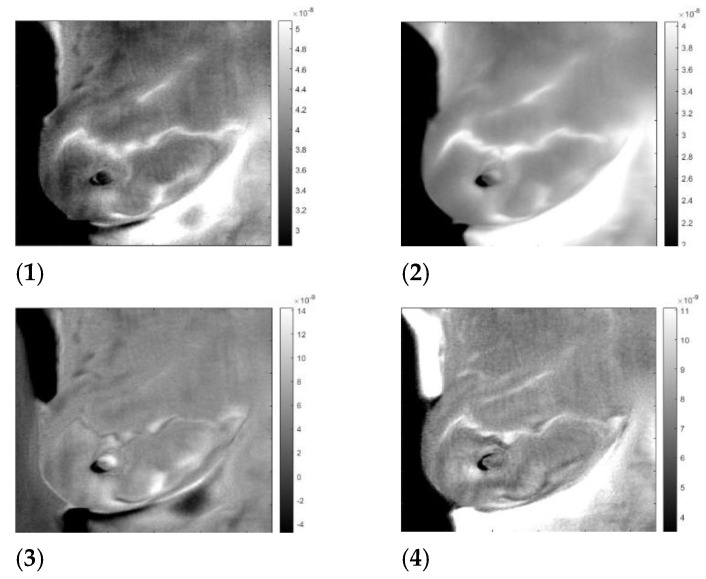
“Sick” left breast of patient #T285. Images 1 to 4 after application of the inverse ICA transform. (**1**) and (**2**) are images of thermal reflection, while (**3**) and (**4**) are images of vasoconstriction.

**Table 1 sensors-23-03012-t001:** Relevant medical information. QSL = Upper Side Quadrant, QIL = Lower Side Quadrant. QIM = Upper Middle Quadrant, None of the patients had mammography.

*Patient*	Age	Exam Details	Diagnosis	Quadrant	Biopsy	Complaint
T281	43	Left	Sick	QSL	left	Pain in both breasts with burning
T282	46	Left	Sick	QIM	left	Pain in both breasts
T285	57	Right	Sick	QIL	right	Pain in the right breast
T286	50	Left	Infiltrating Ductal Carcinoma	QSL	left	Itching in both breasts, secretion in the left breast

## Data Availability

Publicly available datasets were analyzed in this study. This data can be found here: [http://visual.ic.uff.br/dmi/prontuario/home.php].

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
