# Peer review of "Dynamic Vascular Imaging Using Active Breast Thermography"

_sensors, 2023, doi:10.3390/s23063012_

Round 1
Reviewer 1 Report
The authors reported that by irradiating normal and diseased breasts with cold air and analyzing dynamic changes in their far-infrared images, they were able to find different points of temperature change. They claim to have found a different vasoconstriction response in the diseased breast than that seen in the normal breast. I think this method has the potential to demonstrate the clinical utility of IR modality in diagnostic imaging. Therefore, I think it is appropriate to publish it as it is.
However, I think the following minor points need to be fixed.
1) In the section of 3.3 Analysis, despite being written in Arabic numerals in the figure, some numbers are written in alphabetical notation. For example, “figure two” à “figure 2” on line 160, “figure three” à “figure 3” on line 161, “columns one and two” à “columns 1and 2” on line 160.
2) On Line 63, The authors [7,]8--à The authors [7,8]
Author Response
Corrected.
Reviewer 2 Report
This is an intriguing thermal image analysis study that aims to differentiate between healthy and diseased breasts. It is crucial in the study of breast cancer screening.
1. Include the authors' conclusion in the abstract.
2. Line 27 should be corrected. Skin cancer is the most common type of cancer in women.
3. Add a reference to the affirmative: “Early detection is critical for better outcomes.” (line 28). Include bibliographic references to back up all of the statements in the first paragraph.
4. On line 42, insert a reference and inform the sensitivity values.
5. Consider the limitations, sensitivity, specificity, and accuracy of thermography in the diagnosis of breast cancer. Discuss the cost of thermography. Add this to the discussion
6. Lines 45 and 46 should be referenced.
7. Line 53's affirmation contradicts the quoted reference Gautherie (3). Examine the cited author's article as well as the author's other research.
8. Lines 56, 58, 59, and 61 should be referenced.
9. Line 64, explains why Ohashi had limited success.
10. On line 65, use the authors' names instead of (7,8).
11. On line 71, cite the reference.
12. Explain how the authors concluded in line 84 that thermography can detect missed cases by mammography.
13. Put a sentence from line 93 as a limitation of the study in the discussion and explain it better.
14. Please let me know which hormones are on line 108.
15. On line 109, specify the response delay time. How long?
16. Explain the concept of reverse polarity on line 111 further.
17. If possible, include an explanatory diagram (figure) in item 2.3.
18. Improve your definition of diseased breasts. What are the diseases? Do you know the outcome of the biopsy?
19. Improve your definition of healthy breasts. Do you have a normal mammogram? What is the normalcy criterion?
20. What are the patients' ages?
21. Please provide more information about patients T281, T282, T285, and T286. Were the images of these patients taken in the same temperature and humidity conditions?
22. Explain more clearly in the results what the difference is between the normal and sick cases' results/images. It was unclear.
23. Line 183 says the depth has been doubled; was this measured? As?
24. Explain how the cooling is done better in the method so that the experiment can be replicated.
25. Better explain Yousef's proposed use of vasodilation. It is only mentioned in the text without explanation to the reader. Compare your experiment to the authors' experiment.
26. In the conclusion, the authors mention cancer; which type of breast cancer was studied in this article? Early? Lump? Size? Give more information about the tumor.
27. Reduce the conclusion to what the study's authors concluded and leave the findings to the discussion.
28. What is the study's true conclusion?
Author Response
I have addressed most of the issues.
- On line 42, insert a reference and inform the sensitivity values.
This is quite problematic, in particular, due to the majority of studies are based on unreliable data [30]
- Consider the limitations, sensitivity, specificity, and accuracy of thermography in the diagnosis of breast cancer. Discuss the cost of thermography. Add this to the discussion.
-
- Improve your definition of diseased breasts. What are the diseases? Do you know the outcome of the biopsy? OK
Reference [30] discusses the limitation of the study. It was good for mathematical training and a total failure as medical information.
'
Reviewer 3 Report
It is quite an interesting manuscript. The topic of this manuscript falls within the scope of Sensors.
This work focuses on using dynamic infrared breast imaging to identify the thermal response of the stationary vascular system and the physiologic vascular response to a temperature stimulus affected by vasomodulation.
The Authors have presented sufficient data. The appropriate tables and figures have been provided. The article is easy to read and logically structured. The conclusions are consistent with the presented evidence and arguments.
the strength of this paper: very interesting topic; material and methods-the right choice of methodology methods, which were presented incomprehensible way; the obtained results are presented in the form of figures, which are clear and easy to understand; the discussion- supports the results properly and refers to the current literature inappropriate manner; the conclusions- based on the obtained results, they are consistent with evidence and arguments. They address the main question posed.
There are only some comments in the reviewer's opinion that should be taken under consideration by the Author:
1. . Please add the limitations of your study
2. Please add in the introduction section, that thermography may be used not only in the diagnosis of breast cancer but also in the treatment, especially in the estimation of radiation reaction after radiotherapy [doi: 10.3390/ijerph192316085]. This information allows the readers to take a broader look at the role of thermography in breast cancer.
3. In the method section, please give more details on the presented women: #T281, #T285, #T282, #T286,
-age
-what was the size of the tumor,
-what was the scale TNM,
-BMI, did it have an influence on a thermographic image?
4. Please write on the future directions of dynamic thermography [ doi: 10.1016/j.cmpb.2019.105074]
Author Response
Addressed